# OpenReview forum: "Speak Easy: Eliciting Harmful Jailbreaks from LLMs with Simple Interactions"
_ICML.cc/2025/Conference — ICML 2025 poster_

### Official Review · Reviewer_fY99 · 2025-03-11

**Overall Recommendation:** 4

**Summary:**

The paper introduces a new perspective about jailbreak research -- does LLM generate harmful response that is actually actionable and informative? This is a critical perspective, as existing works didn't explore this direction well. To measure this, the paper suggests a new metric called HarmScore, providing a fine-grained assessment. Additionally, the authors examine whether people can elicit harmful behaviors with simple interactions, and based on this perspective they designed a new automatic framework Speak Easy. This can be applied to existing jailbreak tactics, and their extensive experimental results show that they can increase both ASR and HarmScore.

## update after rebuttal
I will keep the rating as I gave the highest rating among the reviewers, and I don't find any critical concerns about the paper.

**Claims And Evidence:**

The motivation for HarmScore is very clear and I buy this motivation. Fine-grained attributes about the jailbreaks are also defined well and studied properly.

**Essential References Not Discussed:**

The paper addressed references well.

**Experimental Designs Or Analyses:**

Experiments are conducted well. They tested on multiple benchmarks and did a lot of analysis about different aspects, including ablations. They tested meaningful base models including GPT-4o, one of the frontier models.

**Methods And Evaluation Criteria:**

Their method, Speak Easy is a new jailbreak method that is based on the simple principle using multi-step attack and multilingual attack. Although each of these strategies can be already addressed in previous works, combining them and using in a single framework is a nice idea. Their new eval metric, HarmScore, also makes sense.

**Other Comments Or Suggestions:**

No comments or suggestions.

**Other Strengths And Weaknesses:**

- Paper is well-written and easy to follow.
- I can't find very meaningful weakness from the paper. Good work!

**Questions For Authors:**

No questions.

**Relation To Broader Scientific Literature:**

Previous works mostly relied on ASR, but suggesting a new idea of evaluating how response is actionable and informative is a good idea. I think this paper is providing a meaningful contribution.

**Theoretical Claims:**

There is no theoretical claim in the paper.

---

> ### Author Rebuttal · Authors · 2025-04-01
>
> Thank you for your encouraging feedback and for recognizing the significance of our contributions. We are grateful that you found our work clearly motivated, supported by extensive experimental results, and offers a critical new perspective on jailbreak attacks. This aligns with our primary goal of grounding jailbreak research in realistic, user-facing scenarios—specifically, examining how simple interactions can lead to harmful outputs that are both actionable and informative.
>
> While no specific revisions were suggested, we will continue to refine the paper with additional experiments and improved clarity. Please do not hesitate to let us know if there are any other ways we can strengthen our work. Thank you again for your evaluation and support!

---

### Official Review · Reviewer_U6FU · 2025-03-11

**Overall Recommendation:** 2

**Summary:**

This proposes HarmScore and a HarmScore-based workflow to elicit harm based from LLMs that are trained in more than one languages. There are several models in the loop to 1. decompose the harmful user instruction to sub questions; 2. models to search which language is most vulnerable for the target LLM. 3. combine results from previous steps. The major contribution is the engineering pipeline.

**Claims And Evidence:**

This paper claims there are three contributions. My big concerns are that they are not novel or the novelty most comes from the engineering of the pipeline, instead of based on the claimed new discovery.

First, the paper claims position HarmScore is an important metric for harm eval.
> We identify actionability and informativeness as key attributes that constitute a harmful jailbreak response.

However, by eyeballing the major result in Figure 4, the new score is pretty much calibrated with the conventional ASR score. I do not see why researchers will instead report HarmScore. It is perhaps a good optimization goal in this pipeline Speak Easy, but it is not a convincing new metric.

> We introduce HARMSCORE, a metric grounded in the aforementioned attributes that aligns competitively with human judgments.

I feel its better frame as Harmscore is helpful to provide useful optimization signal to search for jailbreaks. The current phrasing here is unclear what is the importance of this score.

> We show that SPEAK EASY, a simple multi-step and multilingual jailbreak framework, significantly increases the likelihood of harmful responses in both proprietary and open-source LLMs.

The improved HarmScore or ASR are mostly due that LLMs are more vulnerable on low-resource languages -- which is not novel finding and has been reported widely. This undermines the significance of the finding shown by this work.

**Essential References Not Discussed:**

No

**Experimental Designs Or Analyses:**

The experiment design to choose actionability and informativeness from the rest two criteria is overkill in my opinion. They can just be motivated by examples. Adding the experiments to select do not provide extra strength in justification.

The bigger problem is not to use a different metric than ASR because almost every attackers use their own judge system (judge LLM, humans or heuristics).  ASR itself is not defined by rubrics so it is not useful to argue HarmScore is better than a general metric. You can directly argue HarmScore is the better way to compute ASR.

**Methods And Evaluation Criteria:**

The method and evaluations are standard. They are using common benchmarks.

**Other Comments Or Suggestions:**

N/A

**Other Strengths And Weaknesses:**

Pretty good engineering pipeline. With that being said, I think the paper has more engineering contribution than jailbreak science. If the paper can revise the way they frame what they are doing and what contributions they have made to the society (mostly the industry), I will raise the score.

**Questions For Authors:**

N/A

**Relation To Broader Scientific Literature:**

- LLMs are vulnerable to low-resource languages is a fact that is well known. For example https://arxiv.org/abs/2310.02446. The common practice is to study the robustness of safeguard for English.

**Theoretical Claims:**

N/A

---

> ### Author Rebuttal · Authors · 2025-04-01
>
> Thank you for acknowledging the engineering strength of our work and for your thoughtful suggestions! We appreciate your willingness to reconsider your score based on a clearer explanation of our contributions.
>
> ---
> **1. Framing of contributions**
> > The paper has more engineering contribution than jailbreak science.
>
> Our primary motivation is to demonstrate that average users—without technical expertise—can leverage simple, natural interactions with LLMs to elicit harmful outputs. This reveals a critical yet underexplored vulnerability in safety-aligned LLMs. To reflect realistic usage scenarios, we designed Speak Easy as an engineering pipeline that emulates common multistep, multilingual interactions and shows the effectiveness of jailbreaking through everyday use.
>
> **2. Clarifying the role of HarmScore**
> > HarmScore is pretty calibrated with ASR … It’s better to frame HarmScore as helpful to provide useful optimization signal to search for jailbreaks.
>
> Thank you for this observation. We designed HarmScore to measure the real-world impact of jailbreak responses from the perspective of non-expert users, specifically on how actionable and informative a response is, which aligns with perceived harmfulness.
>
> We acknowledge that HarmScore is often correlated with ASR, and we agree that some degree of calibration is expected—since successful jailbreaks are more likely to result in harmful outputs. However, HarmScore and ASR serve different purposes: ASR captures whether the model complies with a harmful request, while HarmScore evaluates how useful that response is to a malicious actor.
>
> To further clarify this distinction, we demonstrate that ASR and HarmScore may not be calibrated in certain harm categories. We report the by-category scores of the Harmful and Harassment categories below, where HarmScore aligns better with human judgment in Table 3 in the main paper. Here, the scores are evaluated on jailbreak outputs from GPT-4o using the Direct Request + Speak Easy baseline, on the HarmBench dataset.
>
> | Category    | GPT-4o ASR | HarmScore |
> |-------------|------------|-----------|
> | Harmful     | 0.33       | 0.83      |
> | Harassment  | 0.32       | 0.87      |
>
> There is a divergence between ASR and HarmScore in these categories. While some outputs may not be labeled as successful jailbreaks by ASR, HarmScore indicates that they can still be actionable and informative in inducing harm. We appreciate the reviewer’s suggestion, and since ASR and HarmScore serve different purposes and are not always calibrated, we believe HarmScore provides valuable additional granularity. Therefore, we report both metrics in our experiments to offer a holistic view.
>
> > The experiment design to choose actionability and informativeness can just be motivated by examples.
>
> Our initial selection of attributes was indeed inspired by qualitative observations of harmful responses. We opted to conduct a controlled human evaluation to empirically validate our intuition and to ensure rigor in how we identify attributes that matter most for jailbreak harmfulness. We believe that grounding these metrics in human judgment provides added clarity, transparency, and robustness, especially as the field advances toward more interpretable and user-centered safety metrics.
>
> > ASR itself is not defined by rubrics so it is not useful to argue HarmScore is better than a general metric. You can directly argue HarmScore is the better way to compute ASR.
>
> We agree and want to clarify that we do not claim HarmScore is a better metric than ASR, but rather that it captures different and complementary aspects of jailbreak evaluation. As demonstrated in Table 3, each metric excels in different categories. This is why we report both HarmScore and ASR throughout the paper to provide a more holistic view of jailbreak efficacy. We will ensure this distinction is made clearer in the revision.
>
> **3. Clarifying the contribution of Speak Easy**
> > The improved HarmScore or ASR are mostly due that LLMs are more vulnerable on low-resource languages.
>
> We acknowledge that vulnerabilities in low-resource languages have been previously observed. Our motivation is to show that this vulnerability can be effectively leveraged through simple, realistic interactions. Specifically, our work shows that combining multi-step reasoning with multilingual queries results in significantly more harmful outputs, even in safety-aligned models.
>
> In Section 5.4, we show that multi-step interactions contribute more significantly to increased harmfulness, rather than multilinguality alone. Figure 5 shows that higher-resource languages (e.g., English) are selected more frequently in response selection, suggesting that the gains are not solely driven by low-resource language exploitation.
>
> ---
> We hope our responses have addressed your concerns. Please let us know if further clarification is needed. If the issues are resolved, we would be grateful if you could consider raising your score.

---

> > ### Comment · Reviewer_U6FU · 2025-04-06
> >
> > Thanks for the responses from the authors.
> >
> > Overall I think the authors have attempted to address a lot of confusions in the paper presentation. I think these rewrite would strengthen the paper. But still I think the contributions here are important enough for a workshop paper but might be a bit boarderline for the main conference track.  I therefore remain my original score.

---

> > > ### Author Response · Authors · 2025-04-09
> > >
> > > We thank the reviewer for their feedback, and we hope to use this opportunity to clarify the practical significance of our work.
> > >
> > > ---
> > >
> > > Our central contribution is to highlight a previously underexplored attack vector: **the alarming effectiveness with which non-technical users can elicit harmful outputs from LLMs through simple, realistic interactions**, rather than technical jailbreak strategies or sophisticated prompt engineering.
> > >
> > > Most importantly, **our results demonstrate the need for a new type of red-teaming prior to deployment**: instead of evaluating isolated one-shot prompts, it is critical to test realistic user patterns that reflect how LLMs are used in the wild (e.g., multilingual, multi-turn interactions). This has crucial safety implications for model developers; without accounting for these scenarios, current safety evaluations could underestimate real-world vulnerabilities. This is experimentally supported by our results that such vulnerabilities persist across models and benchmarks.
> > >
> > > To further support this perspective, we introduce HarmScore to assess the real-world level of harm that LLM outputs might cause. **To the best of our knowledge, HarmScore is the first to evaluate specific harm-related attributes through controlled human studies**, and to identify actionability and informativeness as key factors for measuring harmfulness.
> > >
> > > ---
> > >
> > > Therefore, we believe our contributions have broad relevance and are well suited for the main conference track. We hope the reviewer will consider the demonstrated significance when evaluating the paper.

---

### Official Review · Reviewer_Qo8S · 2025-03-14

**Overall Recommendation:** 2

**Summary:**

The paper studies jailbreaking LLMs and makes two key contributions: 1) the paper presents a new metric (HarmScore) for jailbreak effectiveness as an alternative to the commonly adopted attack success rate (ASR). The paper first considers four attributes of LLM responses (Informativeness, Conciseness, Actionability, and Coherence) and with correlation analysis against human annotations of response harmfulness, it shows that Informativeness and Actionability have the largest impact on the overall harmfulness of the responses. According, the paper introduces HarmScore which combines both attributes in a single score. 2) the second contribution is arguing for attack strategies that average user would consider for obtaining harmful content from LLMs (e.g., unlike technically involved attacks such as GCG). For that the paper proposed Speak-Easy, a jailbreak method that works as follows: the harmful query is broken down with an LLM prompt into m subqueries, each of which is submitted to the target LLM in n different languages (automatically translated). Then, the fine-tuned a Llama3.1-8B model on synthetic data to serve as a response selector. The paper shows that the presented method boosts both ASR and HarmScore on 4 benchmarks and a 3 LLMs. Furthermore, it shows that the method can be integrated with existing jailbreak methods (GCG and TAP) and provides additional gains in attack effectiveness.

## update after rebuttal
For the comparison between the TAP and SpeakEasy, I understand that the main goal of the paper is more on presenting attacks that the average user would consider for obtaining harmful. The paper/rebuttal does not provide a convincing argument why an average user would not consider a strategy that is simulated by other attacks, e.g., TAP. The paper needs to make that aspect a lot more concrete.

**Claims And Evidence:**

Yes.
1. the paper claims HarmScore better correlates with human judgement for response harmfulness (supported by results in 3)
2. the paper claims Speak-easy boosts ASR/Harmscore (supported by results in figure 4)

**Essential References Not Discussed:**

NA

**Experimental Designs Or Analyses:**

I have the following concerns:

1. Table 2: the paper does not mention what these numbers are. I can guess they are some form of agreement with human annotations. But what kind of agreement score was used.

2. Relatedly, two of the baselines in table 2 (same applies to llama3.1) are off-the-shelf reward models. It is not clear how these models were used to obtained informativeness and actionability scores. The paper needs to provide a specific justification for using reward models for such evaluation.

3. The experiments with GCG-T and TAP-T are very vague: both methods are used in some *T*transfer mode. The paper needs to explain 1) what models were used to generate the attacks, 2) TAP is applicable to all models evaluated. Why did the paper use the Transfer variant of it?

4. In Line 280, the paper gives an unsupported explanation why HarmScore does not perform well on the cybercrime category. The paper needs to provide evidence for that explanation e.g., the percentage of prompts that ask for "content generation rather than actionable guidelines".

**Methods And Evaluation Criteria:**

yes. I have the following concerns:

1. Part of Harmscore definition (line 158) is a binary flag for refusal and the definition is just limiting that to a set of predefined refusal strings S. I think the definition can be made more general and not be limited to a specific implementation detail (e.g., a list of refusal strings).

2. Relatedly, the paper does not provide any info on how that list was curated and how matching was done for the rest of the reported Harmscore results.

3. The response selector was trained on GPT-4 collected data. What kind of scores were collected? Line 254 mentions "output scores" without providing any information about what they look like.

**Other Comments Or Suggestions:**

NA

**Other Strengths And Weaknesses:**

Weakness:

If we compare TAP to Speak-Easy (both starting from the same query), Figure 4 shows that TAP still significantly outperforms Speak-Easy significantly. TAP is essentially a prompt rewriting strategy which is not something we can confidently claim a typical user would not attempt. Compared to the question decomposition and translation to 6 languages (the Speak Easy approach) I would not say they are significantly different in terms of being realistic for a typical user to try.

**Questions For Authors:**

Please see above.

**Relation To Broader Scientific Literature:**

1. The proposed metric is a significant deviation from the commonly used ASR. It can motive further research on customized evaluators for LLM-produced content safety.

2. The paper also argues for attacks that typical LLM users can come up with which is also an interesting deviation for optimization and search-based attacks.

**Theoretical Claims:**

NA

---

> ### Author Rebuttal · Authors · 2025-04-01
>
> Thank you for your detailed review and valuable suggestions! We would like to clarify the experimental details and will ensure these are clearly presented in the revised paper.
>
> ---
> **1. Refusal string in HarmScore**
>
> We use the list of refusal strings from the GCG paper [1] to check whether a response contains any refusal words (L158–160, right column). This serves as an initial filter to decide whether to evaluate actionability and informativeness in HarmScore. Since detecting refusal patterns is not the focus of HarmScore, we adopt this existing list which was previously used in jailbreak evaluations [2].
>
> [1] Zou et al., 2023. Universal and Transferable Adversarial Attacks on Aligned Language Models.
> [2] Robey et al., 2023. SmoothLLM: Defending Large Language Models Against Jailbreaking Attacks.
>
> **2. Regarding response selection models**
> > The response selector was trained on GPT-4 collected data. What kind of scores were collected?
>
> In training the response selection models, two types of scores were collected:
>
> 1. During curating training data: We collect binary labels from GPT-4 on whether a query-response pair is actionable and informative, then construct preference pairs by pairing actionable vs. non-actionable responses (and informative vs. uninformative responses) under the same query.
> 2. During training: We train a Llama3-8B model using iterative DPO to maximize the log-likelihood margin between preferred and non-preferred responses, and the model outputs a continuous score for each attribute. To mitigate extreme values, we apply a sigmoid function and map scores to the $[0,1]$ interval. The “output scores” mentioned in L254 refers to the raw scores produced by the reward model prior to the sigmoid transformation.
>
> Further details are in L212-214 right column and Appendix B2.
>
> > Table 2: the paper does not mention what these numbers are.
>
> The scores are the accuracy in assigning higher scores to the preferred responses (actionable or informative) compared to less preferred ones. The construction of the preference test sets is described in L256-263 left column.
>
> > How are the baseline models in Table 2 selected and how are they used in evaluation?
>
> The selected baseline models are trained to produce a scalar score that captures response quality, and we select them due to their strong performance on RewardBench [3] at the time of our experiments. Actionability and informativeness can be viewed as corollaries of quality and hence we use them for off-the-shelf comparison.
>
> [3] Lambert et al., 2024. RewardBench: Evaluating Reward Models for Language Modeling.
>
> **3. Implementation details of GCG-T and TAP-T**
> > The paper needs to explain 1) what models were used to generate the attacks, 2) TAP is applicable to all models evaluated. Why did the paper use the Transfer variant of it?
>
> We describe each method in Appendix C and will ensure that these details are more clearly communicated in the revised paper.
>
> - For GCG-T, we use the Vicuna-7B and Vicuna-13B models, which is the standard setup in the GCG paper.
>
> - For TAP-T, we use GPT-4o as both judge and target and Mixtral 8x7B as the attack generator. We adopt the transfer variant because combining TAP with Speak Easy could lead to excessive computational requirements (up to 10×10×4×3 queries per harmful query). Additionally, according to HarmBench, TAP-T outperforms TAP on GPT-4 and hence is a more suitable choice for comparison.
>
> **4. Performance comparison of Speak Easy and TAP**
>
> While we agree that TAP outperforms Speak Easy on ASR, the main objective of our paper is not to introduce Speak Easy as a method that displaces existing methods. Instead, we show that simply employing multi-step and multilingual interactions can substantially increase harmfulness, both in a standalone setting and when combined with existing methods. Prompt rewriting strategies such as TAP may also be accessible to typical users, but integrating Speak Easy with TAP increases ASR and HarmScore to a greater extent. Additionally, as shown in Figure 4 and Table 10, Speak Easy yields higher HarmScore than TAP-T, likely because TAP responses often lean toward creative storytelling and lack actionable, informative content.
>
> **5. HarmScore underperforms in cybercrime category**
>
> We conduct an additional experiment to label queries in the cybercrime and misinformation categories (where HarmScore underperforms) in Table 3, with chemical category as a control. We instruct GPT-4o to label each query as either a “content generation” or “actionable guideline” request:
>  - Misinformation: 100% content generation
>  - Cybercrime: 70% content generation
>  - Chemical: 100% actionable guidelines
>
> This mismatch supports our hypothesis that HarmScore struggles to assess response actionability for content generation questions in the cybercrime and misinformation categories.
>
> ---
> We hope these clarifications address your concerns and would appreciate your consideration in raising your score.

---

### Official Review · Reviewer_EZQp · 2025-03-19

**Overall Recommendation:** 3

**Summary:**

This paper investigates vulnerabilities in large language models (LLMs) by demonstrating that harmful jailbreaks can be elicited through simple multi-step and multilingual interactions.
-  First, the authors identify actionability and informativeness as key attributes that constitute a harmful jailbreak response.
- Then, the authors introduce HARMSCORE, a new metric that evaluates the effectiveness of a jailbreak response in enabling harmful actions, and propose SPEAK EASY. The jailbreak framework that exploits common human-LLM interactions to bypass safety guardrails.
- Experimental results across multiple safety-aligned LLMs (including GPT-4o, Qwen2, and Llama-3.3-70B-Instruct) and four jailbreak benchmarks show that SPEAK EASY significantly increases attack success rates (ASR) and HARMSCORE, revealing overlooked vulnerabilities in existing LLM defenses.

**Claims And Evidence:**

- Claim 1: Jailbroken responses that are both actionable and informative are more effective in enabling harmful actions.

   - Evidence: Human evaluations confirm that responses with high actionability and informativeness scores are perceived as more harmful. The authors validate this by introducing HARMSCORE, which aligns well with human judgments.

- Claim 2: Simple multi-step and multilingual interactions can bypass LLM safety mechanisms.
   - Evidence: SPEAK EASY significantly increases ASR and HARMSCORE across different models and benchmarks, showing that decomposing a query into multiple steps and translating it into different languages can evade safety filters.

- Claim 3: HARMSCORE provides a more fine-grained assessment of jailbreak harmfulness compared to ASR.
  - Evidence: HARMSCORE achieves a higher Pearson correlation with human judgments than ASR, particularly for queries that require actionable and informative responses.

- Claim 4: SPEAK EASY can be integrated into existing jailbreak techniques (e.g., GCG-T and TAP-T) to further enhance attack effectiveness.
  - Evidence: When combined with these methods, SPEAK EASY improves ASR by up to 0.48 and HARMSCORE by up to 0.64.

The claims are well-supported by quantitative results and human evaluations, though the generalizability to unseen attack techniques could be explored further.

**Essential References Not Discussed:**

no

**Experimental Designs Or Analyses:**

- The experiments are well-structured, systematically testing the effects of query decomposition, multilingual queries, and response selection strategies.
- The inclusion of ablation studies strengthens the findings, showing the importance of step count and language diversity.

**Methods And Evaluation Criteria:**

- The experimental setup is rigorous, evaluating proprietary and open-source LLMs across four established jailbreak benchmarks.
-  The proposed HARMSCORE metric is well-justified through human evaluations, and the ablation studies effectively isolate key factors contributing to jailbreak success.
- there are some ablation studies to adjust the number of steps, the choice of languages, and the selection of responses at each stage.

**Other Comments Or Suggestions:**

Report ASR and HARMSCORE for each language separately to identify which languages are most effective for jailbreaking.

**Other Strengths And Weaknesses:**

Strengths:
- The paper is well-written and the figures are easy to follow.
- The paper highlights a previously underexplored attack vector—common user interactions—rather than technical jailbreak strategies.
- The findings have significant implications for LLM safety, demonstrating that even non-technical users can elicit harmful responses.
- The inclusion of multiple models, benchmarks, and ablation studies makes the results robust.

Weaknesses:
- The paper does not report ASR and HARMSCORE for individual languages, making it unclear which languages are the most vulnerable to jailbreaks. It focuses on the overall impact of multilingual jailbreaks rather than comparing specific languages’ effectiveness in bypassing safety mechanisms.

**Questions For Authors:**

see weakness

**Relation To Broader Scientific Literature:**

This work builds on prior research on jailbreak attacks, red-teaming, and adversarial prompting.

**Theoretical Claims:**

The authors do not introduce formal theorems, the metric HARMSCORE is empirically validated through human alignment studies.

---

> ### Author Rebuttal · Authors · 2025-04-01
>
> Thank you for your detailed feedback and for acknowledging the significance of our work and its robust experimental results! We appreciate the opportunity to expand on our language evaluation and jailbreak experiments.
>
> ---
> **1. Reporting ASR and HarmScore for individual languages**
> > Report ASR and HARMSCORE for each language separately to identify which languages are most effective for jailbreaking.
>
> We agree that reporting ASR and HarmScore for individual languages would offer valuable insight into language-specific vulnerabilities. In the main paper, we focused on aggregate results because Speak Easy operates over multilingual, multistep interactions, and the final response often comprises content from multiple languages, selected based on informativeness and actionability.
>
> To approximate the contribution of individual languages, we reported results under the “Fixed Language” setting in Appendix C, Table 11, where responses to all subqueries were selected from a single fixed language. We additionally report ASR and HarmScore for each language in this setting below:
>
> | Language         | ASR   | HarmScore | Actionability | Informativeness |
> |------------------|-------|------------|----------------|------------------|
> | English          | 0.370 | 0.477      | 0.440          | 0.568            |
> | Chinese          | 0.435 | 0.447      | 0.425          | 0.552            |
> | Turkish          | 0.350 | 0.456      | 0.406          | 0.588            |
> | Ukrainian        | 0.300 | 0.381      | 0.324          | 0.516            |
> | Thai             | 0.310 | 0.450      | 0.404          | 0.567            |
> | Zulu             | 0.340 | 0.362      | 0.331          | 0.492            |
> | **Speak Easy (6 languages)** | **0.560** | **0.779** | **0.736**        | **0.889**          |
>
> High-resource languages demonstrate greater vulnerabilities, as Chinese has the highest ASR and English has the highest HarmScore. However, using any single language consistently underperforms compared to Speak Easy’s multilingual response selection. This supports our core claim that multilingual querying, when combined with multi-step decomposition, is a key factor in enabling stronger jailbreaks.
>
> **2. Additional Experiments to Demonstrate the Generalizability of Speak Easy**
> > The claims are well-supported by quantitative results and human evaluations, though the generalizability to unseen attack techniques could be explored further.
>
> We appreciate your suggestion to further evaluate the generalizability of Speak Easy to unseen attack strategies. In the main paper, we integrate Speak Easy with two attack paradigms, adversarial suffix optimization (GCG-T) and prompt-based optimization (TAP-T). To extend our analysis, we evaluate Speak Easy with a recent, third class of jailbreak that exploits the generalization gap in safety training by using past-tense phrasing [1]. We first run the baseline Past Tense Attack using GPT-4o on all four benchmarks, with a single attempt per query. To integrate Speak Easy, we use GPT-4o to reformulate the malicious query into past tense, and then apply the standard Speak Easy pipeline.
>
> The table below presents our results. Combining Speak Easy with the Past Tense Attack consistently improves both ASR and HarmScore across all benchmarks, compared to using the attack alone.
>
> |                          | HarmBench      |              | AdvBench       |              | Sorry-Bench    |              | Med-Safety-Bench |              |
> |--------------------------|----------------|--------------|----------------|--------------|----------------|--------------|------------------|--------------|
> |                          | ASR            | HarmScore    | ASR            | HarmScore    | ASR            | HarmScore    | ASR              | HarmScore    |
> | Past Tense Attack        | 0.380          | 0.322        | 0.454          | 0.304        | 0.358          | 0.473        | 0.193            | 0.525        |
> | Past Tense + Speak Easy  | **0.640**      | **0.586**    | **0.702**      | **0.679**    | **0.584**      | **0.721**    | **0.316**        | **0.782**    |
>
> [1] Andriushchenko and Flammarion, 2024. Does Refusal Training in LLMs Generalize to the Past Tense?
>
> ---
> Please let us know if further clarification or results would be helpful. If accepted, will we use the additional page to incorporate the additional experimental results and analysis. If you find our clarifications satisfactory, we would be grateful if you would consider raising your score. Thank you again for your thoughtful feedback!

---

### Decision · Program_Chairs · 2025-05-01

**Decision:**

Accept (poster)

**Comment:**

The paper focuses on jailbreaking attacks in LLMs. The first idea is to understand whether the attacks will actually be useful and correlated with what we as humans would find actionable. To that end, the paper introduces a metric called HarmScore that suggests it should be used in addition to the standard ASR. The paper also explores an attack, called Speak Easy, which is a multi-step multilingual attack. Multilingual attacks have been used in the past; multi-turn attacks have also been exploited in the past, so I do not find much novelty here. Nevertheless, the idea of exploring whether the attack is useful in practice is interesting.